# Interface Design of VR Driverless Vehicle System on User-Prioritized Experience Requirements

**DOI:** 10.3390/s25175341

**Published:** 2025-08-28

**Authors:** Haibin Xia, Yu Zhang, Xuan Li, Dixin Liu, Wanting Wang

**Affiliations:** School of Art, East China Jiaotong University, South Campus, Nanchang 330013, China; xhb@ecjtu.edu.cn (H.X.); wangwt@ecjtu.edu.cn (W.W.)

**Keywords:** user prioritized experience, A-KANO model, VR, driveless vechicle system, interface design

## Abstract

The prioritization of user requirements is neglected in most existing interface designs for driverless vehicle systems, which may incur safety risks, fragmented user experiences, development resource wastage, and weakened market competitiveness. Accordingly, this paper proposes a hybrid interface design method for a virtual reality (VR) driverless vehicle system by combining a A-KANO model and system usability scale (SUS). Firstly, we obtain key words, and a total of 23 demand points are collected through word frequency analysis via combining with user interview and observation method; secondly, 21 demand points are derived from A-KANO model analysis and prioritized for function development; and finally, design practice is carried out according to the ranking results, and virtual reality technology is used to build a VR unmanned vehicle system in order to simulate the interface interaction of a driverless vehicle system. Then, the VR driverless vehicle system is used as a test experimental environment for user evaluation, and combined with the SUS scale to evaluate the user-prioritized experience requirements for practical verification. Empirical results demonstrate that this method effectively categorizes multifaceted user needs, providing actionable solutions to enhance passenger experience and optimize service system design in future autonomous driving scenarios.

## 1. Introduction

The era of rapid advancement in AI technology has coincided with the transformative shift from traditional automobiles to intelligent systems, positioning autonomous driving technology as a critical component of this intelligent transformation. In 2023, China’s autonomous driving market demonstrated vigorous growth, with the market size reaching approximately RMB 330.1 billion, representing a year-on-year increase of 14.1%. Analysts from the China Commercial Industry Research Institute predicted that the autonomous driving market size in China would further rise to RMB 383.2 billion in 2024 [1,2]. Concurrently, the Chinese government has prioritized the development of autonomous vehicles. In December 2023, the Ministry of Transport issued the Guidelines for the Safety Services of Autonomous Vehicles (Trial), which outline operational licensing requirements for autonomous vehicles engaged in urban public transportation, taxi services, and other commercial activities [3]. In August 2024, the Beijing Municipal Bureau of Economy and Information Technology released the Beijing Autonomous Vehicle Regulations (Draft for Comments), clarifying application scenarios, protocols for addressing traffic violations, and liability frameworks for accidents involving autonomous vehicles [4]. These data points and policy initiatives underscore the immense potential and appeal of the autonomous vehicle industry, driving significant capital inflows into the sector and providing robust momentum for technological innovation and industrial upgrading. With the widespread application of autonomous driving technology, user behaviors and demands within driving environments have undergone notable shifts. However, the complex requirements of emerging application scenarios now pose challenges to traditional interaction interfaces. Accordingly, this paper first begins with the prioritization of user experience to conduct their behaviors and requirements. Then, the A-KANO model is employed to categorize and quantitatively analyze requirement points, leading to interface design proposals. Subsequently, these interface design proposals will be integrated using virtual reality (VR) technology to construct a VR-based autonomous vehicle system. Finally, the system usability scale (SUS) is utilized for design evaluation, offering insights for further enhancing the interactive experience of autonomous vehicle system interfaces.

## 2. Related Works

### 2.1. Overview of User-Prioritized Experience in Driverless Vehicle Systems

In this section, we categorize the related works into four parts: an overview of user-prioritized experience in driverless vehicle systems, VR in driverless vehicle systems and the A-KANO-SUS design model. User experience (UX) refers to the perceptions and responses generated by users when using a product, system, or service, encompassing all feelings before, during, and after use [5]. The concept of UX was first proposed by Donald Norman in 1995 and has gradually become an essential research direction in the field of design [6]. The constituent elements of UX are highly complex, involving multiple dimensions. UX extends beyond the direct usage experience of a product or service, encompassing all touchpoints and interaction experiences throughout the product’s lifecycle, including demand definition, product development, brand promotion, and user support. Scholars have proposed a five-level model of UX, including strategic, scope, structure, skeleton, and surface layers, to delineate its different dimensions [7]. In the study of user prioritization methodology, the core of prioritization methods is mainly divided into three types: the hierarchical analysis method (AHP), KANO model, and quality house (QFD). It is mainly used in product design and interaction design, through the construction of models, and as a way to prioritize functions or requirements; it focuses on the highest level of design in the design and development of products. In addition, user profiling and travel graph methods are also used to classify user characteristics and behavioral paths to assist in prioritization decisions. The success of driverless vehicles, as a core breakthrough in the field of intelligent transportation, relies on the dual drive of technological maturity and user acceptance. However, research has shown that users’ trust, safety perception and interaction experience of self-driving systems are key factors in determining their acceptance [8]. The autonomy level of an autonomous driving system (from L0 to L4) significantly affects user acceptance and experience. Rödel et al. found that users’ trust and enjoyment of an autonomous driving system fluctuated with the autonomy level, with L3 systems being more challenging due to the need for the user to take over at all times [9]. In addition, safety requirements differed across autonomy levels, with users more concerned about the reliability of “emergency takeover” in L3 systems [10]. Users’ expectations of driverless vehicles vary according to the type of task and usage scenario. In primary driving tasks, users prefer autonomous Level 2–3 to reduce the operational burden, while in secondary tasks, users rely more on driverless Level 4–5 [11]. This difference suggests that the design needs to optimize the user experience by scenario. To enhance user experience, researchers have proposed user preference-based driving control strategies. Bae et al. developed the occupant preference metric (OPM), which personalizes driving style by defining user tolerance thresholds for acceleration and jerk [12]. The method was validated in simulation to enhance driving comfort. The ease of use of the interaction interface is at the heart of the user experience, and Rajanen points out that user-centered design reduces development costs and enhances brand appeal, especially in L3-level systems, where simplifying the operating logic and enhancing the transparency of information reduces the cognitive burden on the user [13]. In addition, multimodal interactions such as voice and haptic feedback have been shown to enhance user trust in the system [14]. Although UX has been widely applied in driver interaction design for autonomous vehicles to address user experience challenges, multi-level user needs are neglected in implementing UX prioritization methods. Theocharis and Amanatidis systematically explored user interface design for fully autonomous vehicles, including interaction modes (e.g., voice, touch, gesture), interface design principles, subjective user experience metrics (e.g., satisfaction, utility), and the application of multimodal interaction techniques [15]. Some studies have emphasized the importance of emotional design in self-driving cars, suggesting that emotional design can enhance user acceptance and satisfaction with self-driving cars [16].

### 2.2. Overview of VR in Driverless Vehicle Systems

The development of driverless vehicles, as the core of intelligent transportation systems, relies on complex human–machine interaction and safety verification. However, the high cost and risk of real-world road testing has prompted researchers to turn to virtual reality technology. Studies have shown that VR technology can provide highly immersive driving simulation environments for testing the performance and user behavior of autonomous driving systems [17]. VR provides an important tool for the development and evaluation of autonomous driving systems through highly immersive and low-cost simulations. VR technology provides a safe platform for driving behavior research through highly immersive environments. VR-OOM presents a VR-based in-vehicle driving simulator that enables testers to experience an autonomous driving takeover scenario in a controlled environment through a customized scenario design, validating its effectiveness in real driving [18]. This research significantly improves the immersion and comfort of the VR in-vehicle experience by eliminating the visual impact of vehicle rotation [19]. Goedicke et al. developed the “VR-in-Car” system developed to realize the low-cost validation of the interface design of self-driving vehicles by integrating the mobile vehicle and virtual reality technology [20]. With the development of autonomous driving technology, human–machine interaction (HMI) design has become an important direction of research. VR technology provides new solutions in this field to enhance the quality of interaction between users and autonomous driving systems through immersive experiences. Some scholars have studied the optimization of the operator’s situational awareness through VR interface in remote monitoring systems [21]. Another scholar explored the impact of VR on user behavior and alertness when used in a vehicle and found that users with VR headsets showed higher alertness and less fatigue during task execution [22]. This suggests that VR can not only provide entertainment functions, but can also improve users’ attention and operational efficiency to a certain extent. At the same time, the study suggests that VR has also been used to train drivers’ operational skills in takeover request scenarios [23]. In addition, VR can be used to personalize the design of the driving experience. Research has shown that by adjusting the layout, lighting, sound, and other elements of the virtual environment, users can customize the driving environment according to their own needs, thus enhancing overall comfort and satisfaction [24]. This personalization not only helps to improve the user’s driving experience, but also enhances their trust in the autonomous driving system. Although significant progress has been made in VR application for autonomous driving, existing research primarily targets single user groups (e.g., safety testing) rather than systematically comparing need prioritization across demographics like drivers, passengers, and pedestrians—who prioritize system responsiveness, in-vehicle experience, and obstacle avoidance, respectively. Current VR platforms validate isolated functions (e.g., collision detection) but neglect modeling user need hierarchies. We can see that VR offers innovative tools for autonomous system development, but few studies focus on cross-user need prioritization in VR environments.

### 2.3. Overview of the A-KANO-SUS Design Model

The KANO model was proposed by Japanese quality engineer Noriaki Kano in 1984; it is used to analyze and categorize user needs and their impact on product satisfaction [25]. This model classifies user needs into five categories—Must-be Requirements (M), One-dimensional Requirements (O), Attractive Requirements (A), Indifferent Requirements (I), and Reverse Requirements (R)—according to their influence on user satisfaction. The KANO model is a method used to analyze the relationship between product characteristics and consumer satisfaction, assisting enterprises in identifying and prioritizing different types of customer needs during product development to enhance customer satisfaction and product competitiveness. The A-KANO model, proposed by Xu [26], is an effective method for analyzing user needs. Furthermore, Song [27] utilized the A-KANO model to study the functional needs of Chinese users regarding recreational vehicles. Li and Wei [28] employed the A-KANO model to conduct a design study on user needs for sleep aid products, aiming to enhance user satisfaction and alleviate insomnia issues. Liu and Sun [29] introduced the A-Kano model into the design study of female postpartum rehabilitation apps, aiming to improve user satisfaction and alleviate issues stemming from postpartum problems. The system usability scale (SUS) is used to assess users’ subjective satisfaction with the usability of a system or application [30]. The SUS is widely applied in the usability assessment of various systems and applications, such as the development assessment of job information systems and interactive multimedia applications. Through the SUS, designers can gain a better understanding of whether products or services meet user expectations and make improvement decisions accordingly. Wang Yuhui discussed improvements for measurement methods for perceived usability through the establishment of a Chinese version of the SUS, exploring the use of the SUS for integrated modeling and analysis of longitudinal studies of systems and proposing a comprehensive assessment method based on two measurement modes [31]. Tian [32] discussed the use of eye-tracking technology to evaluate and optimize the operational convenience and accuracy of command and control systems, utilizing the NASA-TLX scale and SUS to verify assessment results from the perspectives of human cognitive load and system usability. Therefore, the SUS can be used for comparing schemes and usability scoring, providing robust validation for design.

The A-KANO model is an extension of the KANO model used to analyze the hierarchy of user needs and satisfaction; the A-KANO model provides suggestions for improvement from the perspective of the “importance of needs” of the user, which can tap into the potential excitement of the user and help design innovative features. The SUS scale is a scale used to evaluate the overall satisfaction of users with the system, usually through five questions. The SUS scale is a scale used to assess the overall satisfaction of users with a system, usually through five questions, and it provides quantitative feedback from the perspective of the user’s “actual experience” to ensure that the user can easily get started with the product. The combination of the two approaches creates a group validation mechanism, where the A-KANO model is used to identify the hierarchy of user needs and satisfaction, and the SUS is used to validate that these needs have been met and to assess the effectiveness of the design. It prevents the pursuit of innovation in the product development phase from neglecting basic usability, and also prevents over-focusing on the refinement of basic functionality while neglecting the opportunity to create an experience that exceeds the user’s expectations. In light of this, this paper derives optimized design solutions that meet user needs through the priority sorting of design elements with different sensitivity levels. These are then combined with SUS for design validation, further verifying the rationality of the design structure and enhancing the priority user requirements of autonomous vehicle systems in VR environments.

## 3. Design Requirements in Analyzing Driverless Vehicles

In this section, we first conduct market research and analysis of similar products, employing text mining techniques and semi-structured interviews to deeply explore user prioritized experience needs regarding autonomous vehicles from both online and offline dimensions, completing the construction of word clouds and consolidating priority requirement points. Technological advancements have propelled the development of autonomous driving. Autonomous vehicles are rapidly developing cutting-edge technology products that rely on advanced autonomous driving technologies, including sensors (such as radar, lidar, cameras, etc.), artificial intelligence, and data analysis, aiming to achieve autonomous vehicle operation through advanced technologies, thereby enhancing road safety, reducing traffic congestion, improving traffic efficiency, and providing convenience for individuals unable to drive. Autonomous vehicles are classified into levels ranging from L0 (fully manual driving) to L5 (fully autonomous driving), as defined by SAE International, describing the functional capabilities of autonomous driving at different levels. Currently, autonomous vehicles are at the L4 level of driving functionality [33], representing highly autonomous driving. Within specific operational areas, vehicles do not require drivers to be ready to take over, but outside the geographical fence, drivers may need to take over or switch to manual driving mode. Therefore, developers of autonomous vehicles are still exploring the path towards full L5 autonomous driving.

### 3.1. Market Research and Analysis of the Driverless Vehicle System Interface

User acceptance and trust in autonomous vehicles are crucial factors [34]. Studies have shown that the design of human–machine interfaces (HMIs) plays a significant role in establishing user trust in autonomous vehicles [35]. Therefore, the HMI of autonomous vehicles must adapt to different driving scenarios and user needs. For instance, L3 autonomous vehicles require well-designed information interfaces during takeover processes to enhance user trust. Additionally, the ease of use and interactivity of HMIs are essential factors in enhancing user-prioritized experience. Market research was conducted to extract the characteristics of passenger screens in autonomous vehicles, focusing on well-known and widely operated brands in the autonomous driving and intelligent driving fields, such as Apollo Go, Pony.ai, Tesla, AITO, Xpeng, and NIO. Market research reveals that current autonomous vehicle passenger screens exhibit characteristics such as single functional modules, chaotic color usage, limited interaction forms for manual operation, and weak information visualization. Some brands fail to fully consider passengers’ visual and entertainment needs during travel.

By analyzing the automotive system interfaces of currently popular brands, we can identify their similarities and priorities. For instance, Apollo Go, with the most functional integration, has set up functions such as VR road conditions, SOS assistance, air conditioning control, music, and video in the passenger control interface. Compared to Pony.ai, Apollo Go focuses more on users’ entertainment and comfort experience, while Pony.ai places more emphasis on the optimization of basic functions and features a concise interface. The recently launched Tesla Robotaxi, in addition to basic functions, offers more entertainment options and is committed to minimalist design in its HMI. Most of these brands prioritize multifunctional integration and intelligent multi-mode control for passenger HMI. Therefore, further optimization for designs of autonomous vehicle system interfaces should focus on safety, control functionality, entertainment, and visual interaction design.

### 3.2. Online Review Collection and Text Analysis Based on Python Web Crawler Technology

Existing studies have shown that, in online and virtual environments, due to the reduction and release of moral pressure, stronger unconscious behaviors can emerge. Thus, from a psychological perspective, the anonymity, remote interactivity, and ease of information access in online environments provide users with a relatively relaxed and free space for expression, enabling individuals to more naturally reveal their underlying needs at the unconscious level, free from direct social pressure or immediate feedback. Online comment data are vast and diverse, making the process of comment collection complex, involving data capture, processing, and analysis. Therefore, professional methods and software are required. Web scraping technology is currently one of the most effective ways to achieve fast and efficient data acquisition [36]. Compared to traditional data collection methods, which often rely on manual data collection, which is inefficient and more subjective, web scraping technology automates the collection of large amounts of data from the internet through programming, significantly improving the speed and efficiency of data collection. Additionally, word frequency analysis is a vital tool for text mining [37], involving the statistical analysis of the frequency of occurrence of important words in text to determine research priorities and development trends in a particular field, helping scholars grasp future research trends. This article is based on the collection of online comments on the Jitterbit platform. According to Econometrics, the MAUs of Jitterbit and Shutterbit were 840 and 550 million in March 2024, with a year-on-year growth of 14% and 9%, respectively. This indicates that Jittery is ahead of other platforms in terms of user growth and market share [38]. In addition, Huitun Data’s data show that the age distribution of users on the Jitterbit platform has the highest proportion of users aged 31–40, followed by users aged 18–23 and 24–30, while users aged 41–50 years old and users over 50 years old account for a relatively low percentage [39]. Therefore, ShakeMedia is a platform with mainly young users, and its comments can quickly reflect the high demand for emerging technology products but may underestimate the practical demand of middle-aged and old-aged groups. We searched for keywords such as “driverless”, “radish fast running driverless”, “pony Zhixing driverless”, “Tesla Robotaxi ”, filtered videos with higher engagement and more comments, and collected the user comment data from the driverless car experience videos between 23 and 28 October 2024, amounting 13,108 in total. The corresponding comment extraction processing is illustrated in Algorithm 1.

The ROST ContentMining content mining system developed by the ROST Virtual Learning Team at Wuhan University was utilized to perform word segmentation on the online comments collected under these keyword searches. During preprocessing, it was found that the original word segmentation filter word list of the ROST ContentMining content mining system had good word segmentation effects; however, for unrecognized vocabulary, custom word dictionaries needed to be added and imported to improve word segmentation accuracy. Custom dictionaries added included terms such as “driving,” “Lexus,” “steering wheel,” “brake,” “Askey,” “online car-hailing,” “electric vehicle,” “directly,” “overtaking,” “earning money,” “autonomous driving,” “front-wheel drive vehicle,” “Tesla,” “Apple,” “taxi,” “refrigerator,” “TV,” “Trumpchi,” “autonomous driving,” “why,” “Yu Chengdong,” “press conference,” “tow truck,” “front wheel,” and “rear wheel.” Based on the word segmentation results, word frequency analysis was performed, with keywords screened and sorted. Considering that word segmentation may generate many function words with no specific or practical meaning, such as emojis and interjections, these were deleted during the word segmentation stage. In practical operations, although function words such as emojis and interjections were deleted, some irrelevant data, such as function words, remained. Therefore, the top 100 words were taken as high-frequency words. In addition, user demand is mainly focused on the period of unmanned experience; then, based on the related principle of ride scenario, manual screening is carried out to exclude the residual expression vocabulary, question words, and words that do not match with the ride scenario, such as what to do, driver, tears, the whole country, and don’t understand, etc., and ultimately summarize the 29 high-frequency words, as shown in Table 1.
**Algorithm 1** Douyin Comment Extraction Pipeline1:Define request headers:2:headers←{3:      ’accept’:“application/json, text/plain, */*”,4:      “accept-language”:“zh-CN,zh;q=0.9”,5:      ’referer’: “https://www.douyin.com/?recommend=1”6:      ’user-agent’:“Mozilla/5.0 (Windows NT 10.0; Win64; x64)7:               AppleWebKit/537.36 (KHTML, like Gecko)8:               Chrome/125.0.0.0 Safari/537.36",9:      ’cookie’:“Enter target webpage cookie”10:}11:**function** Spider(url)12:    response←GET(url,headers)13:    response.encoding←’utf-8’14:    **return** JSONParse(response)15:**end function**16:**function** GetTime(ctime)17:    timeArray←LocalTime(ctime)18:    formattedTime←Strftime(timeArray,"%Y.%m.%d")19:    **return** formattedTime20:**end function**21:**function** SaveData(comment)22:    ipLabel←’unknown’23:    **if** ’ip_label’∈comment **then**24:        ipLabel←comment[′ip_label′]25:    **end if**26:    Initialize data dictionary:27:    dataDict←{28:          "userID":comment[′user′][′uid′].strip(),29:          "username":comment[′user′][′nickname′].strip(),30:          "CommentTime":GetTime(comment[′create_time]),31:          "IPLocation":ipLabel,32:          "LikeCount":comment[′digg_count′],33:          "CommentContent":comment[′text′].strip().replace(′\n′,″)34:    }35:           ▹ Save data dictionary to storage (file/database)36:**end function**

Based on Table 1, the keywords “how to handle” and “problem” have frequencies of 122 and 83, respectively, ranking 4th and 11th among the keyword set, which places them in the category of higher frequency keywords. This indicates that users have raised numerous questions about autonomous vehicles. Moreover, we performed word segmentation, removed function words, and sorted the word frequencies for the 233 comments associated with the keywords “how to handle” and “problem” to uncover potential areas of concern. The results are summarized in Table 2, with a total of 17 keywords identified. The focus areas of the online comments corresponding to the keywords “how to handle” and “problem” can be clearly discerned.

### 3.3. User Interviews to Obtain Design Requirements

Based on the text mining results, this study conducted user interviews and observations using the Apollo Go autonomous vehicle as an example to understand users’ concerns regarding autonomous vehicles. During the user interviews, we posed pre-set questions to autonomous vehicle experiencers, regular passengers, and researchers related to automotive system interface design. The questions are summarized in Table 3. The answers provided by the experiencers were then collated and classified as initial data. Meanwhile, online observations were made to record any issues encountered by users during their experiences. A total of 12 autonomous vehicle experiencers were surveyed, including 5 males and 7 females aged 20–40, primarily residing in first-tier and new first-tier cities such as Shanghai, Wuhan, Shenzhen, Chengdu, and Chongqing. The information obtained from the interviews and observations was logically sorted and grouped into categories to summarize the potential demands for autonomous vehicles.

### 3.4. Summarize the Design Requirements

The study integrated the high-frequency keywords collected from user reviews through web crawling with the keywords derived from user interviews and observations. After filtering out functionalities that cannot be optimized within the automotive system interface design, merging duplicate and similar keywords, and reasonably excluding inefficient demand vocabulary, we constructed a user requirement expansion table for the interface design of autonomous vehicle systems, comprising a total of 23 requirement points. The results are summarized in Table 4.

In this paper, this is divided into four dimensions: functional demand, security demand, entertainment demand and visual demand. In the dimension of functional demand, through the keywords of “intelligent” and ‘polite’ and the interview content of AI communication, we firstly analyze how users expect to realize “AI voice assistant function” through the “AI voice assistant function”. Polite and efficient communication, such as temporary drop-off requests, is used to solve communication disputes that may exist in traditional traveling. Second, the analysis of keywords such as “halfway” and “on the road” shows that users strongly need the “temporary drop-off and pick-up function” to cope with changes in the itinerary. In addition, “positional” restrictions (such as rear-row seating and lack of space) give rise to the demand for “seat rotation and adjustment functions” to enhance the space and comfort of the ride. Moreover, interviews on “adjusting air conditioning”, “adjusting backrest angle and massage”, and “emergency office handling” point to the user’s need for personalization of the ride environment, which is expressed in the driverless car’s system interface as “adjusting air conditioning”, “adjusting backrest angle and massage”, and “emergency office handling”. The system interface is specifically represented by the “air conditioning control function”, as well as the “seat massage function” and the “temporary office function” (e.g., document processing) in combination with the commuting scenario. In addition, based on the content of the interviews, the interviewers suggested that the system interface of the driverless car should also have the needs of routes, maps, road conditions, and trip scenes, and therefore the system interface design should include the “map function” and the “3D trip scene function”. Meanwhile, the “time display function” was explicitly mentioned in the interviews, aiming to provide convenient access to time information. For the scenario keywords such as “traffic police” and “driver’s license”, which are manifested in cases where the system cannot recognize the hand gestures of the traffic police, the “Passenger Self-Selection Driving Function” is designed. Finally, through interviews, we learned that different users have different language needs, and in order to serve different user groups (e.g., overseas passengers), we need to include “multi-language mode function” in the system interface.

In the dimension of safety needs, keywords such as “accident” and “loss of control” are directly related to the “emergency assistance function”. Secondly, the keyword “safety” points out that passengers consider more safety-related issues when riding, and it is necessary to popularize knowledge of safe riding, which gives rise to the “passenger information function” to guide passengers to understand the safety situation related to driverless cars. At the same time, through the analysis of the two keywords “encounter” and “traffic jam”, it is specifically manifested in the sudden traffic jam situation on the driving road, thus designing the “remote assistance operation function”. Finally, the word “lost” is analyzed to show that passengers have the risk of losing their belongings during the ride, and the “Reminder and confirmation function for exiting passengers’ belongings” is designed.

In the dimension of entertainment needs, the interview content of “watching video, listening to music, and karaoke” pointed out that passengers would like to have basic entertainment functions such as video playback, listening to songs, and singing. In addition, in the interviews, the interviewers clearly expressed that the existing driverless cars have little entertainment content, and thus need to expand entertainment content, such as game functions, reflecting users’ expectations for diversified in-vehicle entertainment content.

In the dimension of visual requirements, the feedback from user interviews pointed out that “UI texture is simple”, “safety tips are simple”, and “the contrast of layers is weak”, which points to the need to strengthen the contrast between the background and the layers, as well as the need to integrate the safety tips into the UI. The requirements point to strengthening the contrast between the background and layers, and presenting the safety tips in the form of animation to improve the efficiency of information recognition and visual guidance. In addition, comments such as “few wallpaper choices” and “simple 3D scene modeling” point to users’ pursuit of interface aesthetics and customization, and the demand is reflected in the provision of “multi-theme, personalized mode function” and the optimization of the “3D scene modeling” function “and optimize the ”3D scene rendering effect mode”.

## 4. Construction of a Functional Requirement Model Based on the A-KANO Model

As shown in Figure 1, functional service characteristics are first clarified through text analysis and demand attribute extraction; subsequently, the A-KANO model questionnaire is designed and implemented to classify demands based on user feedback, distinguishing must-be, one-dimensional, and other attributes; and finally, sensitivity prioritization quantifies and ranks 21 functions, driving data-informed design optimization decisions. This method systematically transforms user demands into concrete design priorities, providing a scientific basis for core feature development.

### 4.1. Design and Collection of A-KANO Questionnaire

Based on the online questionnaire, a standardized setting was used to explain each function. The types of questions include the respondents’ satisfaction and importance of the passenger control interface in driverless vehicles with this function. The questionnaire is shown in Table 5. The scale has five items: “like”, “granted”, “indifferent”, “reluctantly accept”, “and” “dislike”. In order to reduce the impact of negative answers, asymmetric values are used to measure the degree of user satisfaction or dissatisfaction. According to the user demand elements summarized above, the A-Kano survey questionnaire covers four levels of safety, entertainment, control function and vision, including 23 demand indicators. A total of 162 questionnaires were collected, and invalid questionnaires were removed, with 153 valid questionnaires in total.

### 4.2. A-KANO Data Analysis

In this step, we first calculate the average satisfaction with the function $\bar{h}$, the mean dissatisfaction when the function is absent $\bar{l}$, and the average weight $\bar{p}$. $\bar{h}$, $\bar{l}$, $\bar{p}$ determine the properties of each function. Secondly, if each respondent is *j* and the number of respondents is *n*, then the survey results for each user demand are $\sum_{j}^{n}=\left(h_{j},l_{j},p_{j}\right),\;j=1,2,\ldots ,n$ (*n* = 153), $\bar{h}$, $\bar{l}$ and $\bar{p}$ specific calculation steps are as follows:1.Calculate the average satisfaction rate for a certain function $\bar{h}$:(1)$$\bar{h}=\frac{1}{p}\sum_{j=1}^{n}p_{j}h_{j},$$2.Calculate the mean dissatisfaction without a function $\bar{l}$:(2)$$\bar{l}=\frac{1}{p}\sum_{j=1}^{n}p_{j}l_{j},$$3.Calculate the mean value of a certain function weight $\bar{p}$:(3)$$\bar{p}=\frac{1}{n}\left(p_{1}+p_{2}+\ldots +p_{n}\right).$$

In order to seek the quality weight classification of functional services, the satisfaction of each function can be represented by the vector $S_{i}=\left(s_{i},\alpha_{i}\right)$, where i is the function, and i=1,2,,23. $S_{i}$ represents the value of the two-dimensional satisfaction of the function from the origin to the function, namely the functional importance; $\alpha_{i}$ represents the angle between the function and the X axis, that is, user satisfaction. The calculation formula is as follows:(4)$$S_{i}=\sqrt{{\bar{h}}^{2}+{\bar{l}}^{2}},$$(5)$$\alpha =\arctan(\bar{h}/\bar{l}),$$

The distance interval of $S_{i}$ and the angle range of $\alpha_{i}$ are shown in Table 6.

In the A-KANO model proposed by Qianli Xu, the core of the model is the introduction of KANO Index and KANO Classifier to realize the quantitative analysis of customers’ needs. KANO Index calculates the customer’s “Importance” and “Satisfaction” of each demand through questionnaire survey and data analysis,. The size of the vector ($S_{i}$) is called the Importance index, and the angle ($\alpha_{i}$) is called the Satisfaction index. Importance indicates the degree of customer’s attention to the demand, and Satisfaction indicates the degree of customer’s satisfaction with the demand fulfilment. KANO Classifier is based on the calculation results of KANO Index, and customer needs are divided into four categories: charismatic needs, expected needs, basic needs and undifferentiated needs. Based on the results of data statistics, the vector score interval of Equation (Equation 5) can be divided into demand attributes, on the basis of the distance interval and the angle range of the A-Kano model analysis diagram, as shown in Figure 2. Among them, the basic needs are the minimum requirements that customers think the product or service must have; if these are missing, this will lead to extreme customer dissatisfaction and even if these needs are met, it will not significantly improve customer satisfaction. Desired needs are the ones that the customer expects to have, and if they are met, they will bring a certain degree of satisfaction, but they will not exceed the customer’s expectations. Charismatic needs are the ones that the customer does not have a clear expectation of, but once they are provided, they will bring significantly increased satisfaction, even beyond the customer’s expectations. No difference requirements have no significant impact on customer satisfaction; whether they are met or not, the customer will not have a special reaction. The order of function prioritization is as follows: basic requirements, desired requirements, attractive requirements, and no difference requirements.

Using the calculation formula of A-Kano model, the statistical results of the survey data of various requirements of the driverless vehicle system interface are shown in Table 6.

In the A-KANO model proposed by Qianli Xu, the magnitude of the vector ($S_{i}$) is called the importance index, and the angle ($\alpha_{i}$) is called the satisfaction index. Based on the data statistics results, according to the vector score of Formula (5) which can be divided into demand attributes, on the basis of distance range of $S_{i}$ and $\alpha_{i}$, we drew an A-Kano model analysis diagram, as shown in Figure 2. User demand is grouped into four categories: attractive demand, expected demand, basic demand and no difference demand. The function priority development order is as follows: basic demand, expected demand, attractive demand, and no difference demand [40].

### 4.3. Design and Analysis

As shown in Table 7, the importance of various indicators was further determined by comparing the values based on user requirements within the same attribute category. The order of importance for the requirements of the autonomous vehicle system interface is as follows: B5 > B1 > B11 > B9 > D4 > A2 > A1 > B3 > B4 > A3 > A4 > B7 > B2 > D2 > C1 > D1 > C4 > B6 > C3 > B8 > C2. Firstly, B5 (“temporary pick-up and drop-off functionality”), B1 (time display function), B11 (remote assistance function), B9 (air conditioning control function), D4 (strong contrast between background and layers), A2 (emergency rescue function), A1 (passenger information), and B3 (map function) have the highest priority and are the greatest drivers for enhancing passenger experience. Therefore, these features can be incorporated into the design of the autonomous vehicle system interface. Secondly, B4 (AI voice assistant function), A3 (VR journey scene), A4 (reminder and confirmation of passenger belongings upon alighting), B7 (seat swivel adjustment function), and B2 (multilingual mode) have relatively high priority. The absence of these features may slightly reduce user satisfaction, while their presence can significantly enhance the user-prioritized experience. Thus, flexible decisions can be made regarding their inclusion. Lastly, D2 (multiple themes and personalization mode function), C1 (music listening function), D1 (safety prompts displayed in animation form), C4 (singing function), B6 (seat massage function), C3 (gaming function), B8 (temporary office function), and C2 (video playback function) have relatively low priority. Their absence has a minimal impact on the overall user requirements for the age-friendly bedside table, but they may cater to certain individualized needs. In summary, the priority order of safety-related requirements for the autonomous vehicle system interface is A2 (emergency rescue function) > A1 (passenger information) > A3 (VR journey scene) > A4 (reminder and confirmation of passenger belongings upon alighting). For basic functions, it is B5 (temporary pick-up and drop-off functionality) > B1 (time display function) > B11 (remote assistance function) > B9 (air conditioning control function) > B3 (map function) > B4 (AI voice assistant function) > B7 (seat swivel adjustment function) > B2 (multilingual mode) > B6 (seat massage function) > B8 (temporary office function). For entertainment, it is C1 (music listening function) > C4 (singing function) > C3 (gaming function) > C2 (video playback function). For visual aspects, it is D4 (strong contrast between background and layers) > D2 (multiple themes and personalization mode function) > D1 (safety prompts displayed in animation form).

### 4.4. Functional Requirement Mapping

Based on the demand classification results derived from the Kano model, the functional design of the unmanned vehicle interface must adhere to the principle of differentiated mapping. Basic requirements (must-be attributes) constitute the core product functionality [41]. The absence of these requirements triggers strong dissatisfaction. Essential attributes such as A2 (emergency rescue function) and B3 (map function) must be transformed into zero-tolerance basic features. For instance, emergency rescue can be implemented as a Real-time Collision Automatic Alert System, while the AI Voice Assistant can be mapped to an Instant Multilingual Voice Command Response Module. Performance requirements (one-dimensional attributes) have importance weights that directly influence user satisfaction and form part of the product experience optimization [42]. Expectation attributes like A3 (VR itinerary scenario) and B7 (seat rotation adjustment function) necessitate achieving linear experience gains. For example, developing a Dynamic Road Condition VR Simulation Engine and a Gesture-Controlled Foldable Desktop System can effectively enhance user satisfaction. Excitement requirements (attractive attributes) are key to differentiated competitiveness and can be upgraded into innovative value-added features [43]. Delight attributes such as B6 (seat massage) and C3 (game function) should be designed to deliver interactions that surpass expectations, exemplified by Somatosensory Adaptive Swivel Seats and supporting facilities like an Immersive KTV System. No difference requirements have the lowest priority for resource allocation and represent features to avoid [44]. Attributes like B10 (selection of the driving function) should be prioritized for exclusion to prevent resource wastage.

### 4.5. VR Design Practice

The current design practice scheme is a flat, picture-based display effect. Considering the user-prioritized experience of its unmanned system interface, in the design and evaluation stage, VR technology is utilized to present the unmanned vehicle in a virtualized manner, simulating a relatively real vehicle driving environment for experience. This is in order to enhance the user’s real experience of the unmanned vehicle system interface at the same time as combining it with the SUS scale to carry out user evaluation in order to obtain more accurate assessment data.

This paper focuses on the driverless vehicle system interface, i.e., the passenger interaction screen. In the market research, the passenger interaction screen is mainly placed behind the front seats of the vehicle, and passengers only ride in the second row. The interface design is shown in Figure 3. In view of this characteristic, in the design and construction of the VR environment, the passenger interface module of the virtual unmanned vehicle is mainly interactively realized, and the system interface is set up in two presentation modes. Then, the vehicle driving and interaction interface are linked in order to achieve a real driverless vehicle system interface interaction environment; the system environment is shown in Figure 4.

The VR driverless vehicle system is realized using UE5, which mainly tests the interactive functions of the interface to assist the experiencer in evaluating the system interface, the main test functions of which include route navigation, driving surroundings, audio/video playback, ride instructions, file processing, and other functions. The research object carrier for this study is a driverless vehicle. The current market mainly exists in the form of online vehicles, mainly serving passengers. Therefore, in the VR system design, with reference to the traditional cab passengers’ travel logic, the task scene is designed as a passenger travel taxi scene and the core experience logic is that the passengers enter the vehicle, enter the purpose of the vehicle, the vehicle starts to drive, and the surrounding situation is presented; the experience of the music and other entertainment functions, the experience of office functions, and the end of the trip are assessed. In the simulated driving trip, a voice assistant will be incorporated to assist driving to clarify the purpose of the test experiment.

User needs were obtained through web crawling technology and interview methods, and the A-Kano model was employed to quantify the weight of system interface requirements during the ride experience of autonomous vehicles. Based on the derived priority of these needs, the page layout and richness of functional modules of design elements were reasonably arranged, with visual elements such as color, size, and font thickness used to emphasize the demand factors with higher weight values.

This design optimizes the autonomous vehicle system interface to cater to the diversified usage needs of passengers during the ride. In terms of the safety module, journey safety is a core element in both automotive and interface design. To mitigate personal safety issues arising from program errors or vehicle collisions during the trip, three core functions are provided: emergency assistance, passenger information, and a VR journey scene. In particular, there exist four modules, including a security module, functional module, entertainment functional module and visual design module. Firstly, emergency assistance enables urgent calls for help regarding personal safety issues during the journey, including both audible signals to external individuals and dialing for assistance to traffic police, emergency personnel, and firefighters. Secondly, passenger information educates passengers on relevant riding precautions and safety knowledge. Lastly, the VR journey scene displays real-time surroundings of the vehicle, reducing visual blind spots for rear passengers and enhancing their psychological sense of security. Furthermore, passenger financial security is a crucial factor in enhancing overall security. During the alighting phase, passengers are reminded of forgotten items and asked to confirm retrieval, thereby safeguarding their property. In the basic functional modules, to facilitate route selection or temporary changes during the ride, the map function firstly integrates options for temporary boarding and alighting, VR journey scene road conditions, and route selection. Secondly, to enhance riding comfort, functions for seat rotation adjustment, seat massage, and air conditioning control are provided, allowing adjustment of the seatback angle and cabin environment temperature. For overseas passengers, multiple language options are available to meet the needs of users with language recognition difficulties. Additionally, an AI voice assistant is integrated to provide chat services and voice control operations during the ride, alleviating fear triggered by the empty driver’s seat at night. Moreover, basic office functions such as Word, Excel, PPT, and notes are included in the interface design to accommodate temporary file modifications, along with a time display function to replace the need for checking the time on a mobile phone for convenience. Finally, to address issues such as unrecognized traffic police gestures, vehicle stagnation, and door opening failures during the journey, a remote assistance function is designed to provide remote personnel control, compensating for poor user-prioritized experiences caused by program errors. In the entertainment functional module, the autonomous vehicle system interface integrates entertainment functions such as music listening, karaoke, gaming, and video playback, aiming to alleviate boredom during the ride and enhance the passenger experience. The interface includes apps like NetEase Cloud Music, Bingo Pop, Quanmin K-Song, and iQIYI, allowing users to log in and load their entertainment information. Given that music listening has the highest sensitivity among entertainment modules, it is designed as a widget on the homepage, with other functional widgets presented in a sliding page-turning format following the music listening function. In the visual design module, to highlight UI graphics and ensure easy recognition, the contrast between the background and graphics is firstly enhanced in the graphic design. Simultaneously, the size contrast principle of Gestalt psychology is applied, and graphic design is simplified to avoid the accumulation of visual information and elements. Secondly, personalized themes and layouts, such as bright mode and dark mode, are provided to cater to the specific usage habits and preferences of business people and young users. Lastly, to improve the learning efficiency of safety information prompts, animations are added to the prompts, while maintaining a text format as well. This diversified approach ensures that passengers have a basic understanding of riding information.

### 4.6. Design Evaluation

In industrial usability research, the SUS accounts for 43% of post-study questionnaire usage. Therefore, the SUS was adopted in this study to evaluate product usability. Subsequently, in the subjective evaluation phase, based on the SUS, a questionnaire was designed tailored to the characteristics of the autonomous vehicle system interface, as shown in Table 8.

The scoring participants included six driverless vehicle experiencers and four interface designers aged 16–55 years old, all of whom had conducted 10–15 experiences per month in driverless vehicles of two brands, Radish Express and Pony Smart. Before starting, the scoring participants were firstly introduced to the work, and secondly we formally started to observe and score the participants’ interface operation of the VR driverless vehicle system. The process was repeated for each participant to evaluate the interface design of the system. Finally, the participant ratings were summarized and collated for calculation as follows: For odd-numbered items (items 1/3/5/7/9): “Raw score”-1. For even-numbered items (items 2/4/6/8/10): 5-“Raw score”, where M,$N_{odd}$ is the total SUS score, the odd score for the odd number term and $N_{even}$ even for the even number term.(6)$$M={\left(N_{odd}+N_{even}\right)}^{*}2.5.$$

Lewis and Sauro [45] found that there are primarily two factors influencing SUS data. Items 1, 2, 3, 5, 6, 7, and 8 constitute Factor 1, named “Usability,” with a reliability of 0.91, calculated as the total score of these items. Items 4 and 10 constitute Factor Figure 2, named “Learnability,” with a reliability of 0.7, calculated as the total score of these two items. In order to make the usability and accessibility score compatible with the overall SUS score (both 0–100), the raw data needs to be transformed. Set $M_{o}$ as the availability score; $M_{L}$ is the accessibility score, namely,(7)$$M_{O}=X^{*}3.125.$$(8)$$M_{L}=Y^{*}12.5.$$

As shown in Figure 5, the mean total SUS score for the interface of the autonomous vehicle system is 81.75, and the mean total usability score is 74.375. Based on the SUS scores, the range of acceptability, usability grade and adjective grade was further classified, as shown in Figure 6. The range of acceptability was acceptable, the usability grade was B and the adjective grade was excellent. Therefore, the product used has higher usability.

## 5. Conclusions

This paper investigated the nascent stage of the driverless vehicle market, recognizing the existing lack of full automation and the critical need to address safety and functionality challenges. An optimization approach was developed using the A-KANO-SUS scale within a VR-based driverless vehicle system design. User requirements were first collected via web crawler technology and then used to construct an A-KANO model for function prioritization. Based on this, a VR driverless vehicle system design was formulated and implemented as a test environment. The SUS scale was applied to validate the design, effectively improving the system interface usability and user experience. Findings confirmed the reliability of integrating the A-KANO-SUS scale with Python web crawler technology for unmanned cockpit interface design, demonstrating its practical significance. Nevertheless, relying mainly on jitterbugs and interviews led to a single-sided data source, introducing potential subjectivity. Future studies could leverage multiple platforms for data collection to enhance data objectivity and research reliability.

## Figures and Tables

**Figure 1 sensors-25-05341-f001:**
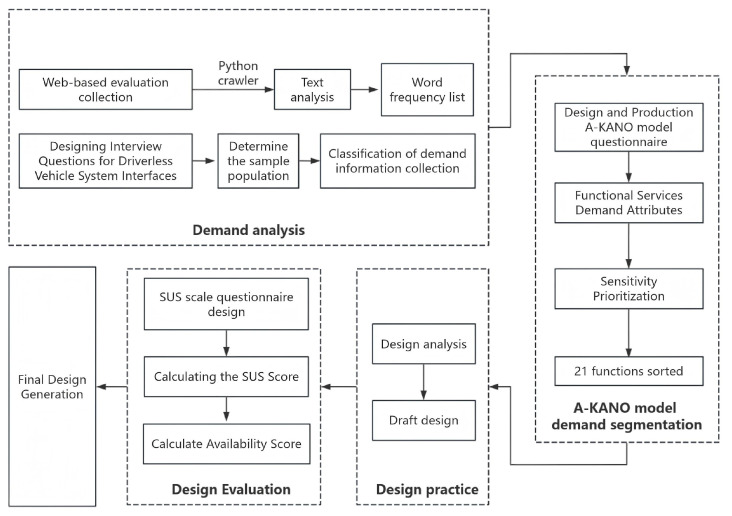
Driverless vehicle interface design program technology path.

**Figure 2 sensors-25-05341-f002:**
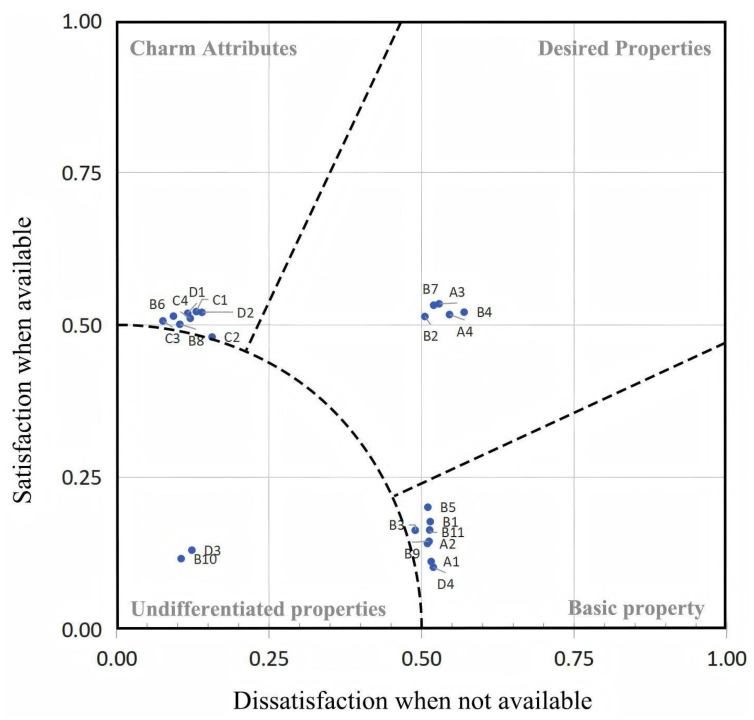
A-KANO model of charismatic, desired, basic, and no difference need analysis chart.

**Figure 3 sensors-25-05341-f003:**
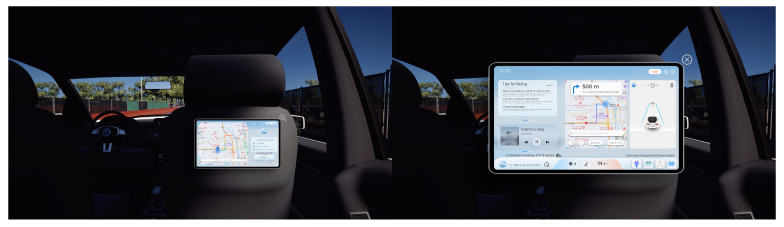
Interface design for unmanned vehicle driving system.

**Figure 4 sensors-25-05341-f004:**
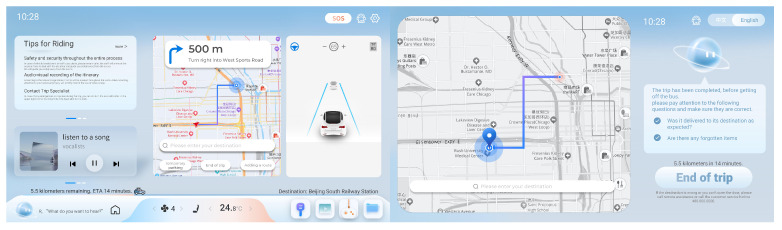
A Diagram of the Virtual Reality Interface for Passenger Interaction in a Driverless Car.

**Figure 5 sensors-25-05341-f005:**
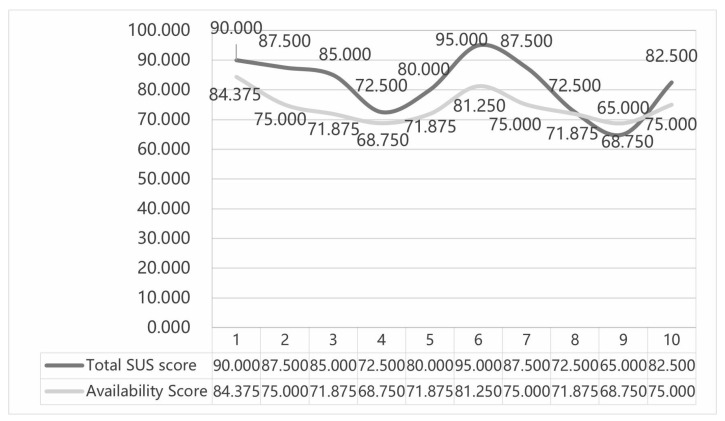
Usability scale scores for interactive interfaces of driverless vehicle systems.

**Figure 6 sensors-25-05341-f006:**
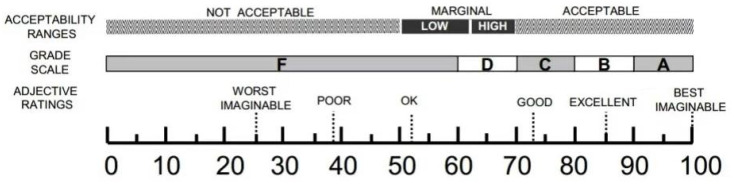
A comparison of the adjective ratings, acceptability ranges, and grade scales, in relation to the average SUS score [46].

**Table 1 sensors-25-05341-t001:** Total word frequency statistics.

Sort	Keyword	Word Frequency	Sort	Keyword	Word Frequency
1	driver	322	16	go-to-work	64
2	science and technology	169	17	evening	60
3	driving license	134	18	intelligence	58
4	how do I	122	19	cellphone	56
5	drive	118	20	on the road	54
6	cheap	114	21	time	54
7	technology	107	22	run into	50
8	steering wheel	101	23	fold	50
9	unmanned	98	24	vision	48
10	safe	92	25	long range	47
11	question	83	26	fear	44
12	popularize	83	27	locality	44
13	dare not	82	28	kilometre	42
14	drive	72	29	on the car	41
15	courtesy	67			

**Table 2 sensors-25-05341-t002:** The frequency statistics of online comments corresponding to “what to do” and “problem”.

Sort	Keyword	Word Frequency
1	on the car	11
2	order	10
3	go	10
4	on the road	9
5	drive	8
6	halfway	8
7	traffic jam	7
8	time	6
9	malfunction	6
10	driving license	5
11	vision	5
12	enjoy together	5
13	hitch	4
14	position	4
15	out of control	4
16	traffic police	4
17	keep in repair	4

**Table 3 sensors-25-05341-t003:** Summary of the interviews and observations.

Order Number	Interview Questions
1	How do you experience the system interface of the driverless vehicle?
2	What functions and features do you think the driverless vehicle system interface should have to better meet your needs?
3	What are the most common problems you encounter when riding in a driverless vehicle?
4	Do you think the operation of the human—computer interaction interface of driverless vehicle is simple and intuitive? Are there some features or options that are hard to find on the interface?
5	Have you ever used entertainment features in a driverless vehicle ride? How is your experience of this feature?
6	Do you think the personalized settings of the human - computer interaction interface of driverless vehicle are enough to meet the needs of users? Do you want to have more personalized options?

**Table 4 sensors-25-05341-t004:** User requirements for the interface design of the driverless car system.

Number	User Requirements	Form
1	AI intelligent voice assistant	Basic Functional Requirements
2	Develop the temporary select pick-up and off function	
3	Seat rotation adjustment function	
4	Temporary office function and adjustment massage function	
5	Self-service selection of the driving function	
6	Map custom route and estimated time function	
7	Map navigation, road conditions function	
8	3D travel scene function	
9	Air conditioning control function	
10	Time display function	
11	Multilingual mode	
12	Remote assistance in the operation function	Security needs
13	Emergency rescue function	
14	Bus instructions function	
15	Passenger item reminder and confirmation function	
16	Video playback function	Entertainment demand
17	Listening function	
18	Singing function	
19	Game function	
20	Safety prompt animation form display	Visual needs
21	Personalized, multi-theme mode function	
22	Graphics strong contrast	
23	The 3D travel scene rendering effect mode	

**Table 5 sensors-25-05341-t005:** Questionnaire design for functional satisfaction and functional importance research based on A-Kano modeling.

Project	Adore	Behoove	Cannot Be Designated as	Unswervingly Accept	Dislike
When this function or service is available	1	0.5	0	−0.25	−0.5
When the function service is not available	−0.5	−0.25	0	0.5	1
How important do you think this feature is?					
It doesn’t matter at all 0 0.1 0.2 0.3 0.4 0.5 0.6 0.7 0.8 0.9 1 very important					

**Table 6 sensors-25-05341-t006:** Requirement attribute classification criteria.

There Is No Differential Demand	Basic Type Requirements	Expected Demand	Charming Demand
S≤0.5	0.5<S≤1	0.5<S≤1	0.5<S≤1
—	α≤25°	25°<α≤65°	65°<α≤90°

**Table 7 sensors-25-05341-t007:** Demand survey data statistics.

Number	User Demand	$\bar{h}$	$\bar{l}$	$\bar{p}$	$s_{i}$	α	Type
A1	Passenger instructions	0.111	0.516	0.677	0.527	12.180	essential attribute
A2	Emergency rescue function	0.141	0.509	0.703	0.529	15.454	essential attribute
A3	VR itinerary scenario	0.535	0.529	0.686	0.752	45.326	Expect attributes
A4	Passenger item reminder and confirmation	0.517	0.546	0.703	0.752	43.459	Expect attributes
B1	Time display function	0.177	0.514	0.690	0.544	18.980	essential attribute
B2	Multilingual mode	0.514	0.505	0.690	0.721	45.478	Expect attributes
B3	map function	0.163	0.489	0.680	0.516	18.378	essential attribute
B4	The AI voice assistant function	0.521	0.570	0.712	0.772	42.443	Expect attributes
B5	Temporary on and off function	0.200	0.510	0.699	0.548	21.456	essential attribute
B6	Seat massage	0.514	0.093	0.690	0.523	79.754	Charm attributes
B7	Seat rotation adjustment function	0.532	0.520	0.692	0.744	45.694	Expect attributes
B8	Temporary office function	0.501	0.103	0.700	0.512	78.335	Charm attributes
B9	Air conditioning control function	0.144	0.512	0.693	0.532	15.752	essential attribute
B10	Passenger self-service selection of the driving function	0.116	0.106	0.556	0.157	47.702	No difference property
B11	Remote assistant function	0.163	0.513	0.707	0.538	17.632	essential attribute
C1	Listening to music function	0.522	0.130	0.703	0.538	75.972	Charm attributes
C2	Video playback function	0.480	0.156	0.692	0.505	71.958	Charm attributes
C3	Game function	0.507	0.076	0.685	0.512	81.505	Charm attributes
C4	Singing function	0.511	0.121	0.708	0.525	76.717	Charm attributes
D1	Safety tips are presented in an animation	0.519	0.116	0.688	0.532	77.368	Charm attributes
D2	Multi-theme, personalized mode function	0.521	0.140	0.703	0.539	74.978	Charm attributes
D3	The VR travel screenerendering effect mode	0.130	0.123	0.533	0.179	46.516	No difference property
D4	Background contrasts strongly with the layers	0.102	0.519	0.702	0.529	11.095	essential attribute

**Table 8 sensors-25-05341-t008:** The SUS scale questionnaire.

Number	Questionnaire Questions	I Don’t Agree ↔ Very Agree				
1	I think I would be willing to use this system more often.	1	2	3	4	5
2	I found that this system is too complicated.	1	2	3	4	5
3	I think this system is very easy to use.	1	2	3	4	5
4	I think I need professional technical support to use the system.	1	2	3	4	5
5	I found that the different functions in this system are well-integrated.	1	2	3	4	5
6	I think there are too many inconsistencies in this system.	1	2	3	4	5
7	I think most people will soon learn to use this system.	1	2	3	4	5
8	I think this system is very cumbersome to use.	1	2	3	4	5
9	I’m very confident in using this system.	1	2	3	4	5
10	I need to learn a lot before using this system.	1	2	3	4	5

## Data Availability

All relevant data are contained within the paper.

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
