# Peer review of "Interface Design of VR Driverless Vehicle System on User-Prioritized Experience Requirements"

_sensors, 2025, doi:10.3390/s25175341_

Round 1

Reviewer 1 Report

Comments and Suggestions for Authors

This paper proposes a hybrid interface design methodology for VR-based driverless vehicle systems by integrating the A-KANO model and the System Usability Scale (SUS). It identifies and prioritizes user experience requirements through text mining, interviews, and surveys, and implements a VR prototype for usability testing. The study aims to enhance user satisfaction and interface functionality in autonomous vehicles. While the approach is novel, the execution and scientific rigor require significant improvement.

1- Figures: The resolution of Figures 1–5 is acceptable, but the captions are minimal and lack descriptive detail. For example, Figure 3 (A-KANO model analysis) should explain the axes and quadrant meanings more clearly.

2- Tables: Tables 1–8 are informative but inconsistently formatted. Table 5 (Satisfaction questionnaire) is particularly confusing and needs clearer labeling and explanation of scoring logic.

3- Missing Elements: Some figures (e.g., Figure 4) are referenced but not visually informative due to lack of annotation or explanation.

4- How were the 23 user requirements in Table 4 validated beyond subjective interviews and word frequency analysis? Please provide a more rigorous justification or triangulation method.

5- Can the authors clarify the rationale for combining the A-KANO model with SUS? How does this hybrid approach improve upon using either method alone?

6- What are the limitations of using Douyin comments as a primary data source for user needs? How do the authors address potential demographic or contextual biases?

7- How was the VR prototype evaluated for realism and fidelity? Was there any validation of the VR environment’s ability to simulate real-world driving scenarios?

8- Can the authors provide statistical significance or confidence intervals for the SUS results? A score of 81.75 is promising, but lacks context without variance or comparative benchmarks.

9- Since fewer than 25% of the references are from after 2020, I strongly recommend the authors update the literature review with more recent studies, especially in the areas of VR usability, autonomous vehicle HMI, and user experience design like the below:

  • https://doi.org/10.3390/futuretransp3010012
  • https://doi.org/10.4108/airo.8026 

Comments on the Quality of English Language

The article needs to double-check for grammaticla issues. Here are some examples of these that should be addressed:

Page 1, Abstract, line 4:

Original: “This paper studies s a hybrid interface design method…”

Correction: “This paper proposes a hybrid interface design method…”

Page 2, Line 41:

Original: “Accordingly, this paper firstly commence from the priority…”

Correction: “Accordingly, this paper first begins with the prioritization…”

Page 6, Line 225:

Original: “at the unconscious level without direct social pressure or immediate feedback.”

Correction: “at an unconscious level, free from direct social pressure or immediate feedback.”

Page 14-15, Line 417-418:

Original: “temporary boarding and alighting function”

Suggestion: Consider rephrasing to “temporary pick-up and drop-off functionality” for clarity and consistency.

Page 18, Line 543:

Original: “the average usability rating is B”

Suggestion: Clarify what “B” means in the context of SUS grading—include a reference or scale.

Reviewer 2 Report

Comments and Suggestions for Authors

The research "Interface Design of VR Driverless Vehicle System on User Prioritized Experience Requirements" stands out for its hybrid methodology that integrates multiple modern techniques to address the research problem. Combining Python web crawlers for large-scale data mining of user comments, the A-KANO model for categorizing and prioritizing user needs, and utilizing a VR environment for immersive, low-cost simulation and evaluation, this approach is robust and novel.

Based on a critical review of the manuscript, I would like the authors to address the specific issues and some areas of improvement:

  1. The manuscript contains significant timeline errors that undermine its credibility. The text states that user comment data was collected from "October 23rd to 28th, 2024" and references a policy from "August 2024".
  2. The study's data collection methods are not sufficiently justified. The online review analysis relied solely on the Douyin platform, without explaining why this single source was chosen or how its user demographics might bias the findings. Although the conclusion acknowledges this as a limitation, the methodology section itself should address this choice.  
  3. The process of refining data lacks transparency and reproducibility as the researchers started with a large volume of online comments, identified the "top 100 words," and then used "manual screening" to arrive at 29 high-frequency words. The specific criteria used for this manual filtering are not explained, making the process subjective and difficult for other researchers to replicate.  
  4. The study employs small sample sizes in its primary research phases without providing adequate justification. Only 12 participants were interviewed, and the final System Usability Scale (SUS) evaluation was performed by only 10; such a small sample size limits the generalizability of the findings, a weakness that should be more prominently discussed.  
  5. There is a significant analytical gap between the initial data and the final design requirements. The manuscript does not clearly explain how the high-frequency keywords from text mining (Table 1) and interview responses were translated into the 23 specific requirement points listed in Table 4. This missing link makes it unclear how the final design requirements were systematically derived from the evidence.  
  6. Grammatical Errors and Awkward Phrasing: The manuscript is marred by numerous grammatical errors and poorly constructed sentences, which hinder readability. For example, sentences like "the complex requirements of emerging application scenarios are now posing challenges to traditional interaction interfaces" require more professional writing.
  7. The paper uses terminology that is occasionally redundant or confusing. [cite_start]For example, in Table 4, the primary category is listed as "Functional requirement needs," which is repetitive. This label is used to group a list of functions that are, by definition, functional requirements.  

Reviewer 3 Report

Comments and Suggestions for Authors

The paper presents a structured methodology for designing a user-centered interface for driverless vehicles, utilizing the A-KANO model to prioritize user needs and the System Usability Scale (SUS) for usability evaluation. The interface is implemented and tested in a Virtual Reality (VR) environment using Unreal Engine 5 (UE5).
The main strengths of this paper, I noticed, are as follows:
• clear problem statement,
• robust methodology,
• user-centric design,
• empirical validation.
Although the topic of the paper is excellent and well structured, there is still room for improvement:
1. Please expand the participant pool by including users across different age groups, cultures, and driving experience levels.
2. Create cross-platform data collection. Please supplement Douyin data with reviews from Reddit, YouTube, or global platforms to diversify insight.
3. Authors can add real-world testing, such as including on-road tests or AR implementations, to bridge VR findings to real environments.
4. Inclusion of developers’ perspectives, such as interface designers' or system engineers’ feedback, could enhance the technical realism of feature feasibility.
I would ask that the authors consider the four points listed above for improvement and include them to the extent possible in the paper.
Expand the literature with current scientific journal articles.
The conclusion must be detailed (which the corrections I propose will also contribute to).
A detailed discussion of the results obtained must be in a separate chapter.
Please check if the authors' email addresses are listed correctly.
Please avoid writing in the first person, and write in the passive voice.

Round 2

Reviewer 1 Report

Comments and Suggestions for Authors

Thanks for providing the replies. I am haooy with this version.